# Commercial Corn Hybrids as a Single Source of Dietary Carotenoids: Effect on Egg Yolk Carotenoid Profile and Pigmentation

Kristina Kljak [ID], Marija Duvnjak [ID], Dalibor Bedeković [ID], Goran Kiš *[ID], Zlatko Janječić [ID] and Darko Grbeša

Faculty of Agriculture, University of Zagreb, Svetošimunska Cesta 25, 10000 Zagreb, Croatia; kkljak@agr.hr (K.K.); mduvnjak@agr.hr (M.D.); dbedekovic@agr.hr (D.B.); zjanjecic@agr.hr (Z.J.); dgrbesa@agr.hr (D.G.)
* Correspondence: kis@agr.hr; Tel.: +385-1-239-3933

**Abstract:** Commercial high-yielding corn hybrids have not been evaluated for their ability to pigment egg yolk. Therefore, the objective of this research was to investigate the effects of commercial hybrids with different carotenoid profiles as the only source of pigments in the diets of hens on yolk color and carotenoid content, as well as the carotenoid deposition efficiency into the yolk. Treatment diets, differing only in one of five corn hybrids, were offered in a completely randomized design in six cages per treatment, each with three hens. Treatment diets and yolks differed in carotenoid profile (contents of lutein, zeaxanthin, β-cryptoxanthin and β-carotene, $p < 0.001$), with total carotenoid contents ranging from 17.13–13.45 μg/g in diet and 25.99–21.97 μg/g in yolk. The treatments differed ($p < 0.001$) in yolk color, which was determined by yolk color fan (10.8–9.83) and CIE Lab (redness; range 12.47–10.05). The highest yolk color intensity was achieved by a diet with the highest content of zeaxanthin, β-cryptoxanthin and β-carotene. The deposition efficiency of lutein and zeaxanthin (25.52 and 26.05%, respectively) was higher than that of β-cryptoxanthin and β-carotene (8.30 and 5.65%, respectively), and the deposition efficiency of all carotenoids decreased with increasing dietary content. Commercial corn hybrids provided adequate yolk color and could be the only source of carotenoids in the diets of hens, which could reduce the cost of egg production and increase farmers' income.

**Keywords:** corn hybrid; egg yolk; lutein; zeaxanthin; β-cryptoxanthin; β-carotene; yolk color; carotenoid deposition efficiency



## 1. Introduction

Producing and providing an adequate amount of high-quality food for the rapidly growing world population requires enormous resources that negatively impact the environment [1]. Among other environmental protection methods, the use of feeds that meet more than one animal requirement shows promising potential by reducing the impact of the production of additional ingredients (such as vitamins and pigments) and contributing to the reduction in the environmental footprint of animal feed production. In this regard, corn grain has the potential to meet poultry energy and resistant starch requirements, as well as contribute to the antioxidant capacity and pigmentation of egg yolk.

The natural color of the yolk is a result of accumulated carotenoids due to hens' ability to deposit carotenoids in the yolk. Carotenoids are compounds with pigmenting but also antioxidant and provitamin A activity, which is why higher yolk color intensity is associated with better laying hen health and better egg quality and flavor [2]. Eggs, in turn, are important natural sources of carotenoids in the human diet, with the xanthophylls lutein and zeaxanthin being prominent due to the prevention and reduction of cataracts and age-related macular degeneration [3,4]. Similar to other animals, hens cannot synthesize carotenoids and must obtain them through diet.

Yolk color is one of the most important attributes for consumer preference in eggs, and one of the most commonly used methods for determining yolk color intensity is the DSM Yolk Color Fan scale (YCF). Yolk color preference varies from country to country, and consumers in Southern Europe prefer a more intense yolk color (YCF > 11), while the majority of consumers worldwide prefer paler eggs [2]. To achieve the desired yolk color, carotenoids are supplied in hen diets as pigment additives, which is an important contribution to the overall cost of egg production. The global carotenoids market reached USD 1.5 billion in 2017 and is expected to reach USD 2.0 billion by 2022, growing at a compound annual growth rate of 5.7% from 2017 to 2022 [5].

A number of factors can influence yolk color, but when considering carotenoids, their content, type and ratio in the diet presents the basis for the possible color [6,7]. In general, the carotenoid profile of the yolk reflects the carotenoid profile of the diet [8], and consequently, the yolk color can be influenced by the addition of different carotenoid sources in the hen diet. It is considered that the standard hen diet, in which the main part is one or more cereals, is insufficient to achieve the desired yolk color without pigment supplementation. These pigments are usually of synthetic origin as they deposit faster and have better efficiency than natural pigments [2,9]. However, when such pigments are added to hen diets, the higher yolk color intensity is not associated with the higher yolk carotenoid content [10]. Furthermore, with the increasing concern regarding healthy nutrition, the synthetic supplementation of food and feed has become an issue. Although numerous natural sources have been investigated over the last three decades, only marigold is commercially available [11].

Corn, which is often the main ingredient of hen diets, is the only cereal with appreciable carotenoid content and an adequate carotenoid profile for egg yolk pigmentation [12]. However, the contribution of corn carotenoids to yolk pigmentation is neglected even in countries with high corn content in hen diets, and pigment supplementation is common in corn-soy-based diets, despite increase of costs and decrease of carotenoid deposition efficiency [6]. In the last decade, biofortification has been used to create corn varieties with enhanced carotenoid content, mainly for human nutrition [13]. When biofortified corn is the only source of carotenoids in hen diets, eggs have higher carotenoid content and more intensive color of yolks compared to non-biofortified corn [14,15]. However, corn varieties biofortified with carotenoids have a much lower yield and a higher price compared to commercial corn hybrids [16], which could limit their use in intensive crop and poultry production.

The most important reason for underestimating the contribution of corn carotenoids to yolk color is likely due to the high genetic variability of carotenoid content in high-yielding hybrids used in commercial production. This genetic variability of corn hybrids suggests that hybrids with high total carotenoid content and adequate profile for yolk pigmentation can be found in the market. At the same time, the use of such hybrids in hen diets does not require pigment addition, which simplifies diet preparation, reduces egg production costs and increases farmers' income. However, commercially available high-yielding corn hybrids have not been evaluated for their ability to pigment the yolk and the efficiency of carotenoid deposition. Therefore, the objective of this research was to investigate the effects of commercial high-yielding hybrids with different carotenoid profiles as the only source of hen diet pigment on yolk color and carotenoid content, and carotenoid deposition efficiency into yolk.

## 2. Materials and Methods

The animal experiment was conducted in accordance with the Croatian directives (Animal Protection Act, OG 102/17, and Regulation on the Protection of Animals Used for Scientific Purposes, OG 55/13), which correspond to the European guidelines for the care and use of animals used for scientific purposes. The animal procedures used in this study were approved by the Bioethics Committee for the protection and welfare of animals

at the University of Zagreb Faculty of Agriculture (KLASA 114-04/17-03/02, URBROJ 251-71-01-17-1).

### 2.1. Corn Hybrids and Treatment Diets

Five commercial and high-yielding yellow corn hybrids (Zea Mays L.) with different carotenoid content (Bc 572, Kekec, Mejaš, Riđan and Pajdaš) were provided by Bc Institute (Zagreb, Croatia). The corn hybrids were grown under the same agro-climate and production conditions. Each hybrid was planted on a 560 m$^2$ test plot located in central Croatia, near Zagreb. At the time of physiological maturity of the grain, each corn hybrid was harvested from the central part of the plot. The harvested grains were dried at 60 °C to 120 g/kg moisture and stored in corn storage bags until the preparation of the experimental diets.

The five experimental diets were formulated to have the same nutrient content and differed only in the corn hybrid. All diets were formulated according to the NRC [17] and updated to be suitable for TETRA SL commercial layer hen hybrid at initial phase of egg production (20–45 weeks of age) [18]; the diet composition and calculated nutrient contents are shown in Table 1. No pigment source other than corn was added. Immediately prior to preparation of the diets, corn grain was ground to pass through a 4-mm screen. Since the objective of the study was to compare commercial high-yielding corn hybrids, the control diet was not included in the experiment.

**Table 1.** Diet composition and calculated nutrient content.

| Ingredient | Content (g/kg) |
|---|---|
| Corn | 600 |
| Soybean meal | 262 |
| Sunflower oil | 30 |
| Calcium carbonate | 88 |
| Monocalcium phosphate | 12 |
| Sodium chloride | 4 |
| DL methionine | 1.5 |
| Vitamin premix [1] | 1.2 |
| TRT Poultry Pack [2] | 1.3 |
| **Calculated nutrient composition** | |
| Crude protein | 170 |
| Crude fat | 55 |
| Crude fibre | 28 |
| Crude ash | 12.7 |
| Calcium | 38 |
| Phosphorus, available | 4.3 |
| Lysine | 8.8 |
| Methionine | 4.2 |
| Metabolic energy (MJ kg$^{-1}$) | 11.6 |

[1] The vitamin premix provided per kg of diet: Vitamin A 10,000 IU, Vitamin D3 2500 IU, Vitamin E 200 mg, Vitamin K3 3 mg, Vitamin B1 1 mg, Vitamin B2 45 mg, Vitamin B3 30 mg, Vitamin B5 10 mg, Vitamin B6 3 mg, Vitamin B7 50 mg, Vitamin B9 0.5 mg, Vitamin B12 25 mg, Choline 400 mg, antioxidant (BHA, EQ) 50 mg. [2] TRT Poultry Pack (Alltech Ireland Ltd., Dunboyne, Ireland) provided per kg of diet: I 1 mg, Fe 5 mg, Cu 5 mg, Mn 30 mg, Zn 30 mg, Se 0.2 mg.

All diets were mixed immediately before the start of the dietary experiment and divided into 5 paper bags. For further analysis, a sample was taken from each bag of the same treatment diet (total of 5 replicates per treatment diet), and samples were stored at −20 °C until carotenoid analysis. Before analysis, corn samples were ground in a laboratory mill (Cyclotec 1093, Foss Tocator, Hoganas, Sweden) equipped with a 0.3-mm screen. All samples were analyzed for dry matter content (DM) determined by drying 3 g of each sample 4 h at 103 °C according to the method ISO 6496:1999 [19].

### 2.2. Hens, Housing and Experimental Design

A total of 90 TETRA-SL 18-week-old laying hens were randomly allotted in groups of 3 to 1 of 30 metal battery cages with 750 cm$^2$ per hen. Diets and water were provided ad libitum to hens. Room temperature was $20 \pm 3$ °C, and the light period consisted of 16 h light per day throughout the experimental period.

After allocating hens to the cages, a 4-week depletion period began. All hens were fed a diet without added pigments and based on barley instead of corn grain and with the same calculated nutrient composition as the experimental diets (Table 1). After depletion, the cages were randomly assigned to one of five dietary treatments (six replicates per dietary treatment). The experimental period lasted 10 weeks. Throughout the experimental period, the number of eggs laid was recorded daily, and diet intake was recorded weekly.

During the experimental period, eggs were collected every three days until the third week to determine stabilization in carotenoid content (i.e., on day 1, 4, 7, 10, 13, 16, 19 and 22 after the beginning of experimental period) and then weekly for color and carotenoid analysis (i.e., on day 28, 35, 42, 49, 56, 63 and 70 after the beginning of experimental period). All eggs were analyzed in the shortest possible time and stored at 4 °C if necessary. Collected eggs were broken immediately before analysis; the yolks were separated from albumen and dried on a paper napkin. After the yolks were separated, their weight was recorded.

### 2.3. Yolk Color Determination

Yolk color was analyzed with both YCF (Egg Multi Tester EMT-5200, Robotmation Co. Ltd., Japan) and Minolta Chroma Meter CR-410 (Minolta Co. Ltd., Osaka, Japan) using the CIE (Commission Internationale d'Eclairage) Lab scale. Color was first determined using the YCF scale; each yolk from eggs collected in the same cage on the same day was analyzed separately, and the average value was taken. Then, the yolks from each cage were combined and mixed carefully to avoid air bubbles, and the color was determined using the CIE Lab scale. The L*, a* and b* values reflect brightness (0 = black, 100 = white), redness ($-a$ = green, a = red) and yellowness ($-b$ = blue, b = yellow), respectively.

### 2.4. Carotenoid Analysis

2.4.1. Experimental Diets

Carotenoids from experimental diets were extracted and quantified according to the procedure described by Kurilich and Juvik [20], using β-apo-carotenal as an internal standard. Each sample was analyzed in triplicate, and the average value was taken as the result. For the extraction, samples were homogenized with ethanol, saponified with 80% KOH and incubated for 10 min at 85 °C in a water bath. The test tubes were then cooled in an ice bath with the addition of deionized water. The carotenoids were extracted with hexane, which was pipetted into a separate tube after centrifugation at 2200× *g* for 10 min (Centric 322A, Tehtnica, Železniki, Slovenia). The extraction procedure was repeated until the colorless upper hexane layer. The collected supernatants were evaporated using vacuum evaporator (Laborata 400 efficient, Heidolph, Schwabach, Germany) and reconstituted in 200 μL acetonitrile:dichloromethane:methanol (45:20:35, *v/v/v*) containing 0.1% BHT.

Carotenoids were separated and quantified using a SpectraSystem HPLC instrument (Thermo Separation Products, Inc., Waltham, MA, USA) equipped with a quaternary gradient pump, an autosampler and a UV/Vis detector. Two sequentially connected C18 reversed-phase columns, Vydac 201TP54 column (5 μm, 4.6 × 150 mm; Hichrom, Reading, UK) and Zorbax RX-C18 column (5 μm, 4.6 × 150 mm; Agilent Technologies, Santa Clara, CA, USA), were used for carotenoid separation. The separation columns were protected by a Supelguard Discovery C18 guard column (5 μm, 4 × 20 mm; Supelco, Bellefonte, PA, USA). The mobile phase consisted of acetonitrile:methanol:dichloromethane (75:25:5, *v/v/v*) containing 0.1% BHT and 0.05% triethylamine. An aliquot of 30 μL was injected, and the



flow rate was 1.8 mL/min. The separations were performed at room temperature, and carotenoids were monitored at 450 nm.

Carotenoids (lutein (purity 99%), zeaxanthin (purity 99%), β-cryptoxanthin (purity 99%) and β-carotene (purity 98%)) were identified by comparing their retention times and quantified by external standardization with calibration curves using commercially available standards (Sigma-Aldrich, Steinheim, Germany; $r^2 \geq 0.99$ for all carotenoids). The total carotenoid content was calculated by summing the contents of the individual carotenoids.

### 2.4.2. Yolks

The changes in carotenoid content in yolks during the experimental period were determined using the reversed-phase HPLC method described previously (Section 2.4.1). Yolks mixed after YCF color analysis were weighed for the carotenoid extraction procedure described by Surai et al. [21] before the determination of color by CIE Lab. Carotenoids were extracted with hexane after yolk homogenization with 2 mL of 5% NaCl aq. solution:ethanol (1:1, *v/v*). Combined extracts were evaporated using a vacuum evaporator and reconstituted in 300 μL acetonitrile:dichloromethane:methanol (45:20:35, *v/v/v*) containing 0.1% BHT.

### 2.5. Carotenoid Deposition Efficiency

The carotenoid deposition efficiency for each cage within dietary treatment was calculated using the following equation [8]:

$$\text{Carotenoid deposition efficiency (\%)} = \text{Carotenoid production by egg/Carotenoid consumption by diet} \times 100$$

where carotenoid production by eggs and consumption by diet were calculated using the following equations:

$$\text{Carotenoid production by egg} = \text{yolk weight (g)} \times \text{yolk carotenoid content (μg/g)} \times \text{egg production (\%)}$$

$$\text{Carotenoid consumption by diet} = \text{diet intake (g/d/hen)} \times \text{diet carotenoid content (μg/g)}$$

based on the data obtained in the hen trial and after sample analysis.

### 2.6. Statistical Analysis

Statistical analyses of the obtained results were performed using SAS statistical software (version 9.4; SAS Institute Inc., Cary, NC, USA). The dietary experiment was conducted as a randomized block design with five dietary treatments, defining a cage with three hens as the experimental unit. Differences between the treatment diets were subjected to an analysis of variance using the MIXED procedure with treatment as the fixed effect. The same procedure was used to analyze differences between treatments in yolk color, carotenoid content and carotenoid deposition efficiency using repeated measurements ANOVA, with results obtained from third week and until end of the dietary experiment. Mean values were defined by the least squares means statement and compared using the PDIFF option; letter groups were determined using the PDMIX macro procedure. The relationship between carotenoid content in treatment diets and yolk color, carotenoid content, and carotenoid deposition efficiency was evaluated using Pearson correlation implemented in the CORR procedure. The threshold for statistical significance was defined as $p < 0.05$.

## 3. Results

### 3.1. Carotenoid Content in Experimental Diets

The experimental diets differed ($p < 0.001$) in the contents of all individual and total carotenoids (Table 2). The average contents of lutein and zeaxanthin in the experimental diets were similar (6.04 and 6.24 μg/g DM, respectively), although the treatments had a wide range of both carotenoids in the experimental diets. Treatment Bc 572 had the lowest dietary lutein content, while the content in the Riđan treatment was twice as high.

Opposite to lutein, the Bc 572 treatment had the highest dietary zeaxanthin content, which was twice as high as in the Riđan and Pajdaš treatments. Treatment Bc 572 also had the highest dietary contents of β-cryptoxanthin and β-carotene, resulting in the highest dietary content of total carotenoids among the treatments. Treatment Bc 572 had a 27% higher total carotenoid content than the treatment with the lowest content, Kekec.

**Table 2.** Average carotenoid content (μg/g DM) in treatment diets differing in corn hybrids.

| Carotenoid | Dietary Treatment | | | | | SEM | *p* |
| | Bc 572 | Kekec | Mejaš | Riđan | Pajdaš | | |
|---|---|---|---|---|---|---|---|
| Lutein | 3.72 [e] | 5.64 [d] | 6.53 [c] | 7.36 [a] | 6.94 [b] | 0.09 | <0.001 |
| Zeaxanthin | 9.99 [a] | 5.87 [b] | 5.78 [b] | 4.57 [d] | 4.98 [c] | 0.10 | <0.001 |
| β-cryptoxanthin | 1.67 [a] | 0.83 [c] | 1.02 [b] | 0.75 [c] | 0.73 [c] | 0.05 | <0.001 |
| β-carotene | 1.74 [a] | 1.12 [c] | 1.45 [b] | 1.16 [c] | 1.40 [b] | 0.07 | <0.001 |
| Total carotenoids | 17.13 [a] | 13.45 [d] | 14.78 [b] | 13.86 [cd] | 14.04 [c] | 0.16 | <0.001 |

Values in a row with different letters differ significantly ($p < 0.05$).

### 3.2. Yolk Color

The yolks from hens fed the tested dietary treatments differed ($p < 0.001$) in color determined by the YCF scale (Table 3), the difference being due to the differentiation of treatment Bc 572 from the other treatments. Despite the differences in the dietary carotenoid profile, the Kekec, Mejaš, Riđan and Pajdaš treatments yielded similar YCF scores. Of the CIE Lab color space parameters, only redness differed between the dietary treatments tested ($p < 0.001$; Table 2), and it followed the same relationship between treatments as the YCF scale. Dietary treatments tended to differ in yellowness ($p = 0.065$), with the lowest value detected in the treatment with the highest redness value, Bc 572.

**Table 3.** Average egg yolk color according to the Yolk Color Fan (YCF) scale and CIE Lab in eggs laid by hens fed dietary treatments differing in corn hybrids.

| Carotenoid | Dietary Treatment | | | | | SEM | *p* |
| | Bc 572 | Kekec | Mejaš | Riđan | Pajdaš | | |
|---|---|---|---|---|---|---|---|
| YCF | 10.8 [a] | 9.92 [b] | 9.96 [b] | 9.98 [b] | 9.83 [b] | 0.08 | <0.001 |
| CIE Lab | | | | | | | |
| L* | 67.27 | 67.64 | 67.76 | 67.70 | 67.35 | 0.21 | 0.351 |
| a* | 12.47 [a] | 11.01 [b] | 11.02 [b] | 11.04 [b] | 10.05 [b] | 0.12 | <0.001 |
| b* | 67.47 | 68.14 | 68.30 | 68.62 | 68.22 | 0.28 | 0.065 |

Values in a row with different letters differ significantly ($p < 0.05$).

The yolk YCF score and the redness and yellowness according to CIE Lab correlated with the content of individual and total carotenoids in the experimental diets. An increase in lutein content in the experimental diets was associated with a decrease in redness and YCF score (r = −0.73 and −0.76, respectively, $p < 0.001$) and an increase in yellowness (r = 0.43, $p < 0.05$). On the other hand, zeaxanthin, β-cryptoxanthin and β-carotene showed an opposite correlation compared to lutein. Zeaxanthin and β-cryptoxanthin correlated positively with redness and YCF (r = 0.77 and 0.78 for zeaxanthin and r = 0.70 and 0.71 for β-cryptoxanthin, respectively, $p < 0.001$) and negatively with yellowness (r = −0.42 and −0.46, respectively, $p < 0.05$). β-carotene and total carotenoids correlated with redness and YCF (r = 0.42 and 0.50 for β-carotene, $p < 0.05$, and r = 0.68 and 0.69 for total carotenoids, $p < 0.001$, respectively).

### 3.3. Yolk Carotenoid Profile

The content of individual and total carotenoids in yolks increased in all treatments during the first two weeks of the experiment (Figure 1). After the second week, the content

of individual carotenoids showed weekly fluctuations, which were more pronounced for β-cryptoxanthin and β-carotene than for lutein and zeaxanthin. Weekly fluctuations in the total yolk carotenoid content were less evident compared to individual carotenoids. No significant decrease in the yolk content of the determined carotenoids was observed by the end of the experiment.

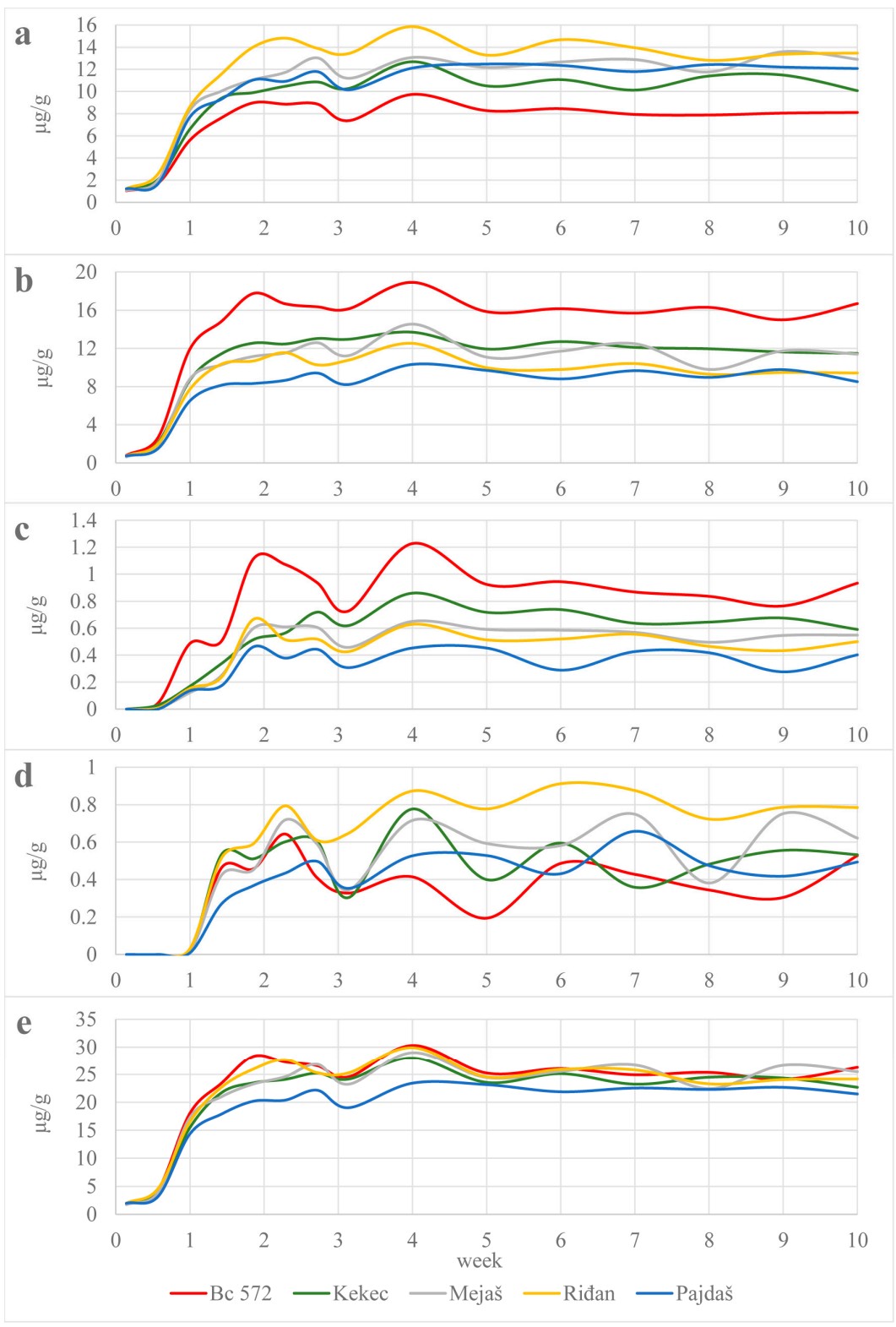

**Figure 1.** Changes in yolk contents of lutein (**a**), zeaxanthin (**b**), β-cryptoxanthin (**c**), β-carotene (**d**) and total carotenoids (**e**) in eggs laid by hens fed dietary treatments differing in corn hybrid.

The yolks of the experimental treatments differed in the content of individual and total carotenoids ($p < 0.001$; Table 4). The average content of lutein and zeaxanthin in the yolks was 11.51 and 12.00 µg/g, and their content was up to 25 times higher than the content of β-cryptoxanthin and β-carotene, on average, 0.61 and 0.56 µg/g, respectively. Treatment Bc 572 had the lowest lutein content and the highest content of zeaxanthin, β-cryptoxanthin and total carotenoids. Treatment Riđan had the highest contents of lutein and β-carotene. The contents of lutein, zeaxanthin and β-cryptoxanthin increased with their increase in the experimental diets (r = 0.93, 0.93 and 0.83, respectively, $p < 0.001$). On the other hand, the yolk content of β-carotene decreased with its content in the experimental diets (r = −0.61, $p < 0.001$).

**Table 4.** Average yolk carotenoid content (µg/g) in eggs laid by hens fed dietary treatments differing in corn hybrids.

| Carotenoid | Dietary Treatment | | | | | SEM | p |
| --- | --- | --- | --- | --- | --- | --- | --- |
| | Bc 572 | Kekec | Mejaš | Riđan | Pajdaš | | |
| Lutein | 8.31 [e] | 10.92 [d] | 12.52 [b] | 13.91 [a] | 11.88 [c] | 0.19 | <0.001 |
| Zeaxanthin | 16.36 [a] | 12.35 [b] | 11.78 [b] | 10.29 [c] | 9.22 [d] | 0.22 | <0.001 |
| β-cryptoxanthin | 0.92 [a] | 0.69 [b] | 0.56 [c] | 0.51 [d] | 0.39 [e] | 0.02 | <0.001 |
| β-carotene | 0.40 [d] | 0.51 [c] | 0.60 [b] | 0.79 [a] | 0.48 [cd] | 0.03 | <0.001 |
| Total carotenoids | 25.99 [a] | 24.47 [b] | 25.47 [ab] | 25.49 [ab] | 21.97 [c] | 0.40 | <0.001 |

Values in a row with different letters differ significantly ($p < 0.05$).

### 3.4. Carotenoid Deposition Efficiency in Yolks

The deposition efficiency of lutein and zeaxanthin (on average, 25.52 and 26.05%, respectively) in the yolks of the tested treatments was up to 10 times higher than the deposition efficiency of β-cryptoxanthin and β-carotene (on average, 8.30 and 5.65%, respectively). The tested treatments differed in the deposition efficiency of all individual and total carotenoids ($p < 0.001$; Table 5). Treatment Pajdaš achieved the lowest values of deposition efficiency for all individual carotenoids except β-carotene, and the deposition efficiency of total carotenoids was similar to the result of treatment Bc 572, despite the high total carotenoid content in the experimental diet of treatment Bc 572. In addition, treatments Kekec, Mejaš and Riđan resulted in a similar deposition efficiency of total carotenoids, regardless of the differences in the carotenoid profile of the experimental diets.

**Table 5.** Average carotenoid deposition efficiency (%) in yolks from eggs laid by hens fed dietary treatments differing in corn hybrids.

| Carotenoid | Dietary Treatment | | | | | SEM | p |
| --- | --- | --- | --- | --- | --- | --- | --- |
| | Bc 572 | Kekec | Mejaš | Riđan | Pajdaš | | |
| Lutein | 28.46 [a] | 26.37 [b] | 26.19 [b] | 24.81 [b] | 21.75 [d] | 0.60 | <0.001 |
| Zeaxanthin | 20.87 [d] | 28.62 [ab] | 27.76 [b] | 29.53 [a] | 23.49 [b] | 0.60 | <0.001 |
| β-cryptoxanthin | 7.00 [c] | 11.21 [a] | 7.52 [c] | 8.87 [b] | 6.90 [c] | 0.32 | <0.001 |
| β-carotene | 2.94 [d] | 6.23 [b] | 5.68 [b] | 8.95 [a] | 4.45 [c] | 0.35 | <0.001 |
| Total carotenoids | 19.31 [b] | 24.74 [a] | 23.50 [a] | 24.15 [a] | 19.85 [b] | 0.50 | <0.001 |

Values in a row with different letters differ significantly ($p < 0.05$).

The deposition efficiency of all individual and total carotenoids correlated negatively with their content in the experimental diets (lutein: r = −0.62, $p < 0.001$; zeaxanthin: r = −0.68, $p < 0.001$; β-cryptoxanthin: r = −0.43, $p < 0.05$; β-carotene: r = −0.79, $p < 0.001$; total carotenoids: r = −0.63, $p < 0.001$).

### 4. Discussion

The only source of carotenoids in the experimental diets was corn; therefore, lutein, zeaxanthin, β-cryptoxanthin and β-carotene, as the main carotenoids in corn [20], were

found in the diets. Consequently, the differences between the experimental diets in the carotenoid profiles were attributed to the differences in the carotenoid profiles of the commercial corn hybrids used. Comparing the experimental diets in terms of their carotenoid profile with other studies is somewhat difficult because the current research focuses on biofortified corn. In those studies, commercial yellow corn has typically been used as the control to which biofortified corn has been compared. Experimental diets prepared with commercial yellow corn in these studies have lower levels of total carotenoids than the diets in the present study. Liu et al. [14] reported that commercial yellow corn contained 24.63 nmol/g, i.e., 13.89 µg/g of total carotenoids, and the resulting diet containing 600 g/kg of corn corresponds to a total carotenoid content of 8.33 µg/g. In addition, Moreno et al. [15] reported that the experimental diet containing 620.6 g/kg of corn had 9.2 µg/g of total carotenoids, while Ortiz et al. [22] reported 5.7 µg/g of total carotenoids in a diet containing 565 g/kg of commercial yellow corn.

Although studies have shown that biofortification generally results in increased carotenoid content in the experimental hen diets, this increase could represent an increase in the content of some specific carotenoids, which, consequently, does not result in an increase in total carotenoids. The experimental diets in the present study had higher contents of lutein, zeaxanthin, β-cryptoxanthin and β-carotene but did not contain violaxanthin and astaxanthin, which were present in the diets with corn fortified with ketocarotenoids in the study by Moreno et al. [15], resulting in similar total carotenoid contents (13.81 µg/g vs. 14.65 µg/g on average in the present study). Similarly, high β-cryptoxanthin corn in the study by Liu et al. [14] provided a similar amount of β-cryptoxanthin and β-carotene (4.71 and 5.31 nmol/g for corn varieties, or 1.56 and 1.71 µg/g, calculated on the basis of corn proportion in the diet, respectively) as treatment Bc 572 in the present study but with a lower total carotenoid amount (13.89 µg/g, calculated on the basis of corn proportion in the diet, vs. 17.13 µg/g, respectively). Therefore, some existing commercial corn hybrids have comparable levels of individual or total carotenoids to carotenoid-biofortified corn, i.e., they could provide their similar levels to the hen through the diet. This implies that some high-yielding commercial corn hybrids, depending on yolk pigmenting ability, could be used in targeted production as corn for laying hens.

The next step in evaluating the suitability of commercial corn hybrids as a single pigment in hen diets was to determine their yolk pigmentation ability. The results of the YCF values obtained show that treatment Bc 572 was the only commercial hybrid able to pigment the yolk to the color acceptable in most markets, while the other treatments pigment the yolk in such way that the color is acceptable in markets where a paler yolk color is desired. Similar values to treatments tested in the present study were achieved by the diet containing 620.6 g/kg of genetically engineered corn fortified with carotenoids in the study by Moreno et al. [15], while the diet containing 565 g/kg of Orange Corn, non-GMO corn biofortified to increase the total and provitamin A carotenoid content in the study by Ortiz et al. [22], achieved even lower values (9 ± 0.3). Diets prepared with the yellow commercial corn from these two studies (YCF scores ~ 6 and 4, respectively) achieved a color significantly lower than the YCF scores of the present study. When considering the CIE Lab, the increase in yolk YCF scores is accompanied by the increase in yolk redness (r = 0.96, $p < 0.001$), suggesting that carotenoids affecting redness contribute most to the changes in the YCF scale. The dietary treatments in the present study resulted in eggs that varied in yolk redness from 10.23 to 12.47, which is higher than the values for biofortified corn in studies by Liu et al. [14] and Ortiz et al. [22] (~5 and 4, respectively).

The commercial corn hybrids in the present study achieved yolk colors comparable to various pigments in other studies. Skřivan et al. [23] reported a similar YCF score (10.55) and redness values (11.51) of yolks from hens fed diets based on 350 g/kg of corn and 280 g/kg wheat supplemented with 950 µg/g of marigold flower extract, while treatment Bc 572 in the present study achieved similar yolk YCF score (10.55) as hens fed diets containing 600 g/kg of wheat supplemented with 15 µg/g of spirulina in a study by Zahroojian et al. [24]. However, the yolk color intensity in the present study was not

as high as that of the commercial synthetic pigment containing canthaxanthin added to the standard diet at a concentration of 8 mg/kg (YCF 13.47, a* 18.76) [10]. The higher pigmentation effect of canthaxanthin compared to the carotenoids present in corn is due to its higher deposition efficiency in the yolk; 37 to 50% of the ingested canthaxanthin is deposited in the yolk [2]. However, although the dietary treatments were not as successful as the synthetic pigments in achieving high YCF scores, the eggs from the present study provide more carotenoids in the human diet, especially lutein and zeaxanthin, at no additional cost to the pigment source. On the other hand, the YCF scores obtained were close to the value of 10.7 obtained in the study by Englmaierová et al. [25], who used a combination of synthetic carotenoids with 2 mg/kg of canthaxanthin and 1.5 mg/kg of ethyl ester of β-apo-8′-carotenoic acid in hen diet.

Based on the correlations between the dietary content of individual carotenoids and color scores, dietary lutein contributed to yolk yellowness, which decreased the YCF score. On the other hand, zeaxanthin, β-cryptoxanthin and β-carotene contributed to yolk redness, which increased the YCF score. These correlations suggest that an adequate profile of commercial corn hybrids for desirable yolk pigmentation has elevated levels of the latter three carotenoids, as found in treatment Bc 572. This observation is in agreement with Liu et al. [14] and Ortiz et al. [22], who also showed that increased zeaxanthin and β-cryptoxanthin in diets as a result of increased content in corn grain resulted in higher yolk color intensity.

The accumulation of carotenoids in the yolks occurred after the fourth day, after the experimental diets were offered to the hens, and saturation was reached after the second week. The duration of the accumulation phase in the present study falls within the range reported in studies with biofortified corn. Moreno et al. [15] reported that saturation was reached after the third week in all dietary treatments, while Ortiz et al. [22] showed that maximum carotenoid accumulation was reached on the 12th day of feeding with dietary treatments. During the experimental period, the yolk levels of lutein and zeaxanthin were higher and more constant than those of β-cryptoxanthin and β-carotene. Moreover, the weekly fluctuations of β-carotene in the yolk showed a wide range in all the treatments tested. These fluctuations most likely reflect the utilization of β-cryptoxanthin and β-carotene by the hen; as previously reported [26], they are converted to vitamin A, with β-carotene being more readily converted than β-cryptoxanthin [14]. As yolk carotenoid levels remained constant until the end of the experiment, commercial corn hybrids provided a stable source of carotenoids for yolk pigmentation.

In agreement with previous reports [8,14], yolk carotenoid profiles of the tested treatments reflected the carotenoid profile of the experimental diets, with zeaxanthin dominating in the eggs of treatments Bc 572 and Kekec and lutein in the eggs of treatments Mejaš, Riđan and Pajdaš. The high correlation coefficients between the dietary and yolk contents of lutein, zeaxanthin and β-cryptoxanthin confirm these considerations. On the other hand, the β-carotene content decreased with the dietary content, in accordance with the preferential conversion to vitamin A. The experimental diets had the same content of ingredients and the other ingredients except corn were from the same batch, thus unifying the effect of the diet matrix on carotenoid bioavailability. From compounds present in the experimental diets, oleic acid was the most important for carotenoids; corn contains 26.8% while sunflower oil contains 20% of this fatty acid [27]. Monounsaturated fatty acids improve the absorption of polar carotenoids, especially when associated with high ME density (11.6 MJ/kg) [28].

The yolk total xanthophyll content reached values in the range of 21.5 to 25.5 µg/g, which was close to lower values in yolks from ecological eggs (20.5–33.5 µg/g) but close to higher values for yolks from free range (12.6–25.5 µg/g), barn (12.1–26.8 µg/g) or eggs (7.3–29.3 µg/g) purchased in local German supermarkets [29]. This comparison suggests that commercial corn hybrids provide more carotenoids than most free-range, barn and cage eggs, in which canthaxanthin, β-apo-8′-ethyl ester and citranaxanthin were found in considerable amounts. Moreover, the yolks from hens fed corn hybrids in the present

study provided more xanthophylls than conventional cage, cage free, cage-free and free-range/pasture and free-range/pasture-organic eggs (21.5, 22.1, 18.2, 18.8, and 22.6 µg/g, respectively) purchased from the local market in the study by Ortiz et al. [22]. The yolks from the later study had YCF scores between 6 and 8, implying that commercial corn hybrids could result in higher color intensity with only a small increase in xanthophyll content in the yolks.

Compared to biofortified corn, commercial corn hybrids were within the range of total carotenoid contents in yolk reported in previous studies. While Liu et al. [14] reported lower values for high-β-carotene and high-β-cryptoxanthin corn (23.61 and 25.86 nmol/g, i.e., 13.37 and 14.61 µg/g, respectively), Ortiz et al. [22] reported a higher value for Orange Corn (29.4 µg/g). Moreno et al. [15] reported results in the freeze-dried yolks and based on the fact that the yolk contains approximately 50% moisture [30], corn enriched in carotenoids resulted in similar contents of total carotenoids (57.5 µg/g of freeze-dried yolk), while corn enriched in ketocarotenoids resulted in twice lower content (26.18 µg/g of freeze-dried yolk). In these previous studies, the biofortification of corn in the diet of hens resulted in increased zeaxanthin and β-cryptoxanthin content and color intensity in the yolks, and a similar result was found with treatment Bc 572 in the present study.

The deposition efficiencies of lutein (on average 25.52%) and zeaxanthin (on average 26.05%) were higher than those of β-cryptoxanthin (on average 8.30%) and β-carotene (on average 5.65%). In addition to the preferential utilization of β-cryptoxanthin and β-carotene as vitamin A, these results also reflect the easier absorption of the more polar carotenoids lutein and zeaxanthin [31]. Furthermore, van het Hof [32] has shown that lutein is five times more bioavailable than β-carotene in humans. The deposition efficiencies of individual carotenoids were within the range of values reported in previous studies [8,33,34]. These studies reported variable deposition efficiencies but also differed in the carotenoid source and its inclusion level in the diet. For example, Karadas et al. [8] reported deposition efficiencies of zeaxanthin ranging from 9.1–30.3% and β-carotene ranging from 0.4–3.8% for diets containing 20 g/kg of lucerne extract, marigold extract and tomato powder. Hammershøj et al. [29] reported values in the range of 18.8–27.4 for lutein, 18.9–27.9 for zeaxanthin and 0.3–1.0 for β-carotene in diets supplemented with 70 g/day/hen of different carrot varieties. Thus, the values reported in the present study show good deposition efficiency of corn carotenoids, including β-carotene, into the egg yolk.

In addition to the differences between individual carotenoids, the deposition efficiency of corn carotenoids also varied with their concentration—the levels of all individual and total carotenoids decreased with increasing dietary content. This relationship between the deposition efficiency into yolk and dietary content has been reported previously for lutein [8,34]. It seems that other corn carotenoids follow the same relationship, in agreement with their decreasing bioaccessibility from the matrix with increasing content in the grain [35]. However, the tested corn hybrids as the only source of carotenoids in hen diets resulted in high yolk carotenoid content and color intensity for the majority of world markets, despite the observed relationship between the deposition efficiency and the dietary content of carotenoids.

## 5. Conclusions

Commercial high-yielding corn hybrids may differ in carotenoid content, but the present study showed that those with high carotenoid contents could be the only source of pigments for yolk pigmentation in hen diets. More so, the dietary treatments resulted in the high yolk content of lutein and zeaxanthin, comparable to biofortified corn, which has been widely studied recently, with no decrease in color intensity compared to synthetic pigment sources widely used in poultry production. Corn hybrids containing higher levels of zeaxanthin and β-cryptoxanthin even resulted in higher yolk color intensity (higher YCF score and redness values). The results of the study suggest that more attention should be given to the selection of existing commercial corn hybrids for poultry; the inclusion of these hybrids in the diet could lead to a reduction in egg production costs.

**Author Contributions:** Conceptualization, K.K. and D.G.; methodology, K.K., M.D. and Z.J.; validation, K.K., D.G. and Z.J.; formal analysis, K.K., M.D., G.K. and D.B.; investigation, K.K., M.D., G.K., D.B. and Z.J.; resources, D.G. and Z.J.; data curation, K.K.; writing—original draft preparation, K.K.; writing—review and editing, D.G., M.D., G.K., D.B. and Z.J.; visualization, K.K.; supervision, D.G.; project administration, K.K. and D.G.; funding acquisition, D.G. All authors have read and agreed to the published version of the manuscript.

**Funding:** This research received no external funding.

**Institutional Review Board Statement:** The study was conducted according to the European guidelines for the care and use of animals used for scientific purposes and approved by the Bioethics Committee for the protection and welfare of animals at the University of Zagreb Faculty of Agriculture (KLASA 114-04/17-03/02, URBROJ 251-71-01-17-1; date of approval 6 June 2017).

**Informed Consent Statement:** Not applicable.

**Data Availability Statement:** The data presented in this study are available on request from the corresponding author.

**Conflicts of Interest:** The authors declare no conflict of interest. The funders had no role in the design of the study; in the collection, analyses, or interpretation of data; in the writing of the manuscript, or in the decision to publish the results.

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
