# Peer review of "Commercial Corn Hybrids as a Single Source of Dietary Carotenoids: Effect on Egg Yolk Carotenoid Profile and Pigmentation"

_sustainability, doi:10.3390/su132112287_

Round 1
Reviewer 1 Report
Dear Authors,
although the paper is interesting and overall well written, in my opinion a major flaw is represented by the absence of a control group managed under the same conditions in order to gain a comparative view on the individual differences between carotenoids present in commercial corn towards biofortified hybrids, in accordance with other recent references. I have highlighted in yellow directly into the text attached below my suggestions and doubts.
Best regards.

Author Response
Attached is a summary of the responses to the comments of reviewers. My coauthors and I appreciated the Reviewers’ constructive comments; we are thankful for the efforts and time regarding the manuscript. We have addressed each of their concerns as outlined below. Numbered lines are from the current revised manuscript. We hope that the revised version of the manuscript will meet Reviewer’s standards.

Reviewer 2 Report
Dear ,
In my opinion the manuscript is well written. It presents a novel approach to the topic.
Author Response
We appreciate the time and effort the Reviewer put into reviewing the manuscript. Also, we are grateful for recognizing our effort to use commercial corn hybrids in sustainable hen nutrition.
Reviewer 3 Report
1- English should improve by a native person. The paper suffers from a poor English structure throughout and cannot be published or reviewed properly in the current format. The manuscript requires a thorough proofread by a native person whose first language is English. The instances of the problem are numerous and this reviewer cannot individually mention them. It is the responsibility of the author(s) to present their work in an acceptable format. Unless the paper is in a reasonable format, it should not have been submitted.
2- The novelty of the study needs to be highlighted compare to other similar studies.
3- Discussion is weak. The discussion needs enhancement with real explanations not only agreements and disagreements. Authors should improve it by the demonstration of biochemical/physiological causes of obtained results. Instead of just justifying results, results should be interpreted, explained to appropriately elaborate inferences. discussion seems to be poor, didn't give good explanations of the results obtained. I think that it must be really improved. Where possible please discuss potential mechanisms behind your observations. You should also expand the links with prior publications in the area, but try to be careful to not over-reach. For the latter, you should highlight potential areas of future study.
4- The scientific background of the topic is poor. In "Introduction" and "Discussion", the authors should cite recent references between 2015-2021 from JCR journals (with impact factor) about recent achievements on the subject. For example, authors should cite to:
Shahsavari K. (2015). Influences of different sources of natural pigments on the color and quality of eggs from hens fed a wheat‐based diet. Iranian J. Appl. Anim. Sci. 5(1), 167-172.
Ortiz, D., Lawson, T., Jarrett, R., Ring, A., Scoles, K. L., Hoverman, L., ... & Rocheford, T. (2021). Biofortified orange corn increases xanthophyll density and yolk pigmentation in egg yolks from laying hens. Poultry Science, 100(7), 101117.5- A detailed "Conclusion" should be provided to state the final result that the authors have reached. Please note you only need to place your conclusion and not keep putting results, because these have already been presented in the manuscript.
6- Author(s) should re-format the references based on journal format. See the instructions for authors.
Author Response
Attached is a summary of the responses to the comments of reviewers. My coauthors and I appreciated the Reviewers’ constructive comments; we are thankful for the efforts and time regarding the manuscript. We have addressed each of their concerns as outlined below. Numbered lines are from the current revised manuscript. We hope that the revised version of the manuscript will meet Reviewer’s standards

Round 2
Reviewer 1 Report
Dear Authors,
I am glad to see that you have accepted the reviewers' suggestions thus improving the manuscript.
Best regards